# Network-level changes in the brain underlie fear memory strength

**Josue Haubrich[1,2]\*, Karim Nader[1]**

[1]Department of Psychology, McGill University, Montréal, Canada; [2]Department of Neurophysiology, Ruhr-University Bochum, Bochum, Germany

**Abstract** The strength of a fear memory significantly influences whether it drives adaptive or maladaptive behavior in the future. Yet, how mild and strong fear memories differ in underlying biology is not well understood. We hypothesized that this distinction may not be exclusively the result of changes within specific brain regions, but rather the outcome of collective changes in connectivity across multiple regions within the neural network. To test this, rats were fear conditioned in protocols of varying intensities to generate mild or strong memories. Neuronal activation driven by recall was measured using c-fos immunohistochemistry in 12 brain regions implicated in fear learning and memory. The interregional coordinated brain activity was computed and graph-based functional networks were generated to compare how mild and strong fear memories differ at the systems level. Our results show that mild fear recall is supported by a well-connected brain network with small-world properties in which the amygdala is well-positioned to be modulated by other regions. In contrast, this connectivity is disrupted in strong fear memories and the amygdala is isolated from other regions. These findings indicate that the neural systems underlying mild and strong fear memories differ, with implications for understanding and treating disorders of fear dysregulation.

## eLife assessment

This **important** study provides **convincing** data in support of the conclusion that weak but not strong fear memories are more easily modified using behavioral and pharmacological approaches potentially as a result of differential connectivity with the amygdala showing greater connectivity through the brain in weak compared to strong memories. The scope of the paper would be strengthened if both sexes were examined and more varied definitions of weak vs. strong memories were used. This paper is of interest to behavioral and neuroscience researchers studying learning, memory, and/or neural networks.

**\*For correspondence:**
Josue.Haubrich@ruhr-uni-bochum.de

**Competing interest:** The authors declare that no competing interests exist.

## Introduction

Whether a fear memory leads to adaptive or maladaptive behavior can be determined by the intensity of the initial aversive experience. A deeper understanding of fear memory requires identifying changes in the neural mechanisms engaged by fear memories of varying levels of aversiveness. Research on rodent fear conditioning has been mostly conducted in mild protocols and comparisons with strong protocols are scarce. While this approach advances our understanding of normal fear memories, it leaves open questions about specific mechanisms in disorders of fear dysregulation, such as post-traumatic stress disorder (PTSD) (*Flores et al., 2018*).

Normal fear memories are critical to survival. These memories trigger appropriate responses that are cue-specific and can be attenuated by extinction (*Fanselow, 2018*; *Homan et al., 2019*; *Alexandra Kredlow et al., 2022*). These adaptive representations are flexible and their content may be

**eLife digest** Remembering the fear that arose during a dangerous experience is important as it teaches us to avoid similar circumstances in the future. The intensity of the initial experience will often influence the strength of the memory. Milder memories often lead to responses that protect individuals from harm (known as adaptive behaviors). However, stronger memories of more traumatic experiences can sometimes trigger disproportionate responses to a situation (known as maladaptive behaviors), such as in individuals with phobias or post-traumatic stress disorder (PTSD).

Forming and retrieving fear memories requires different parts of the brain to work together and send signals to one another. At the core of this network is the amygdala (also known as the fear center of the brain), which other brain regions then feed into to modulate the fear response to ensure it is appropriate and manageable. However, it remained unclear whether neurons in these brain regions wire together differently when recalling mild or more severe fear memories. Identifying these differences may help explain why certain fear memories lead to adaptive behaviors, while others result in maladaptive ones.

To investigate this question, Haubrich and Nader generated fear memories in rats that triggered either mild fear responses or strong responses akin to trauma. Imaging tools were then used to measure the activity and connections between neurons across 12 regions of the brain known to be involved in remembering fearful experiences.

This revealed that recalling mild fear memories resulted in a well-coordinated network of neurons which could effectively send information between the different brain regions. In contrast, severe fear memories led to disrupted overall connectivity, with the amygdala becoming disconnected from the other brain regions.

The results reveal stark contrasts in the pattern of neuronal connections formed by mild and severe fear memories. Investigating the specific pathways involved in these differences will allow scientists to gain a better understanding of why memories of traumatic experiences can lead to maladaptive behaviors, including those formed as a result of PTSD.

updated upon recall through reconsolidation (*Arellano Pérez et al., 2020*; *Forcato et al., 2010*; *Haubrich et al., 2015*; *Monfils et al., 2009*). Importantly, these fear memories can be disrupted by pharmacological interventions targeting their reconsolidation (*Haubrich et al., 2020a*; *Haubrich et al., 2017*; *Nader et al., 2000*; *Pigeon et al., 2022*), offering a wealth of clinical possibilities (*Phelps and Hofmann, 2019*). However, severe aversive events can lead to the formation of strong, maladaptive fear memories that trigger disproportionate and generalized fear responses (*Baldi et al., 2004*; *Corchs and Schiller, 2019*; *Morey et al., 2015*) and tend to be resistant to extinction (*Sangha et al., 2020*) and to undergo reconsolidation (*Haubrich and Nader, 2018a*; *Holehonnur et al., 2016*; *Kindt, 2018*; *Wang et al., 2009*).

The formation and expression of fear memories are believed to rely on the coordinated activity of a network of brain structures. The amygdala is an integral structure in all aspects of fear memory formation and expression (*Sears et al., 2014*; *Zhang et al., 2021*). Subsequent to severe fear learning, it has been shown that changes take place in the composition of glutamate receptors in the amygdala that are linked to maladaptive memories (*Holehonnur et al., 2016*; *Wang et al., 2009*; *Conoscenti et al., 2022*; *Haubrich et al., 2020b*). In the case of contextual fear memories, the involvement of the hippocampus also becomes critical given its role in the processing of context-related information (*Frankland et al., 2019*; *Suzuki et al., 2004*; *Topolnik and Tamboli, 2022*). Other regions, such as the retrosplenial, infralimbic, prelimbic, and cingulate cortices, and the reuniens and paraventricular nuclei of the thalamus, also play important roles in modulating fear memories (*Alexandra Kredlow et al., 2022*; *Barchiesi et al., 2022*; *de Oliveira Alvares and Do-Monte, 2021*; *Levy and Schiller, 2021*; *Milton, 2019*). Importantly, the action of stress hormones and the locus coeruleus-noradrenaline system plays a key role in the formation of maladaptive-like fear memories (*Haubrich et al., 2020b*; *Dębiec et al., 2011*; *Gazarini et al., 2014*). It is not clear how the coordinated activity between these regions changes with the severity of fear memories but basic and clinical findings suggest it may be dysregulated (*Ressler et al., 2022*).

The distinctions between adaptive and maladaptive memories may not be solely attributed to changes within isolated structures or altered connectivity between pairs of structures, but rather to collective changes across multiple structures within the neural network. Combining the quantification of immediate-early genes expression linked to neural activation such as c-fos (*Terstege and Epp, 2023*) and network-based graph analysis (*Bassett et al., 2018*; *Bullmore and Sporns, 2009*; *Roland et al., 2023*), it is possible to reveal the brain functional connectivity during active cognitive experiences, such as memory recall (*Silva et al., 2019*; *Takeuchi et al., 2022*; *Vetere et al., 2017*; *Wheeler et al., 2013*). In network models of neural systems, brain regions are represented as nodes, and functional connections between nodes are represented as edges. Multiple measures can be computed to inform the relationship between nodes and the global characteristics of the network, which can elucidate changes in information processing across different conditions.

In this study, we aimed to shed light on the system mechanisms underlying the recall of mild and strong fear memories. First, we established the behavioral differences triggered by fear conditioning protocols of varying intensities. We then employed c-fos immunohistochemistry to measure brain activation in 12 brain areas critical for fear learning and memory during the recall of mild and strong fear. Subsequently, the interregional coordinated brain activity was computed and functional networks were generated. Our findings indicate that the system mechanisms of fear memories vary based on their intensity as a mild fear recall is supported by a well-connected network while the recall of strong fear is associated with disruptions in network connectivity.

## Results

### The effect of varying fear conditioning intensity on behavioral responses

In order to investigate fear memories of different intensities, animals underwent contextual fear conditioning (CFC) with either 2 (2S) or 10 foot shocks (10S). A control group, in which no shock was administered, was included to provide a baseline behavior (NS). After training, a series of behavioral tests were conducted to determine the characteristics of the resulting memory. One day after training, we assessed the intensity of fear responses triggered by the conditioned context (Test A), and on the following day, fear generalization in a novel context (Test B). In addition, changes in exploratory behavior were assessed in an open-field exploration test conducted 3 d after training.

During training (*Figure 1A*, middle), repeated-measures ANOVA indicated a significant effect of group ($F_{2,19}$ = 255.2, p<0.001), bin ($F_{11,209}$ = 130.4, p<0.001), and group × bin interaction ($F_{22,209}$ = 43.77, p<0.001). Tukey's post hoc revealed that in the 2S group, freezing levels during the concluding 4 min surpassed all previous time bins (p<0.001). In the 10S group, the first 3 min exhibited lower freezing relative to all other bins (p<0.05), and the last 6 min did not differ from each other (p>0.05). This indicates that freezing behavior increased after shock presentations and showed no decline toward the session's end.

The results of Test A and Test B showed that freezing behavior increased as a function of the training intensity (*Figure 1A*, bottom left). Repeated-measures ANOVA indicated a significant effect of group ($F_{2,21}$ = 166, p<0.001) and group × session interaction ($F_{2,21}$ = 12.86, p<0.001). Tukey's post hoc indicated that in context A all groups differed from each other (p<0.001), with the 10S displaying the higher freezing levels. During the test in context B, there was no difference between NS and 2S groups (p=0.31) while the 10S group was higher than all others (p<0.001). To compare fear generalization between animals trained with 2 shocks and 10 shocks, a discrimination index was computed and compared, and a Student's *t*-test revealed that there was a significant difference between groups (*Figure 1A*, bottom center; t(14) = –2.3, p=0.037).

In the open-field test conducted 1 d after test 2 (*Figure 1A*, bottom right), there were no group differences in the time spent in corners ($F_{2,24}$ = 0.987, p=0.4) and total distance moved ($F_{2,24}$ = 1.52, p=0.1). There were also no differences in the time spent in the center ($F_{2,24}$ = 0.302, p=0.7), number of quadrant crossings ($F_{2,24}$ = 1.2, p=0.3), number of crossings on the center ($F_{2,24}$ = 0.301, p=0.7), and average speed ($F_{2,24}$ = 1.56, p=0.2) (data not shown). Thus, suggesting that differences in freezing performance are due to alterations in memory expression and not in motor abilities.

We then evaluated the effect of training intensity on fear extinction (*Figure 1B*). Repeated-measures ANOVA indicated a significant effect between groups ($F_{1,14}$ = 6.67, p=0.02), bin ($F_{6,84}$ =

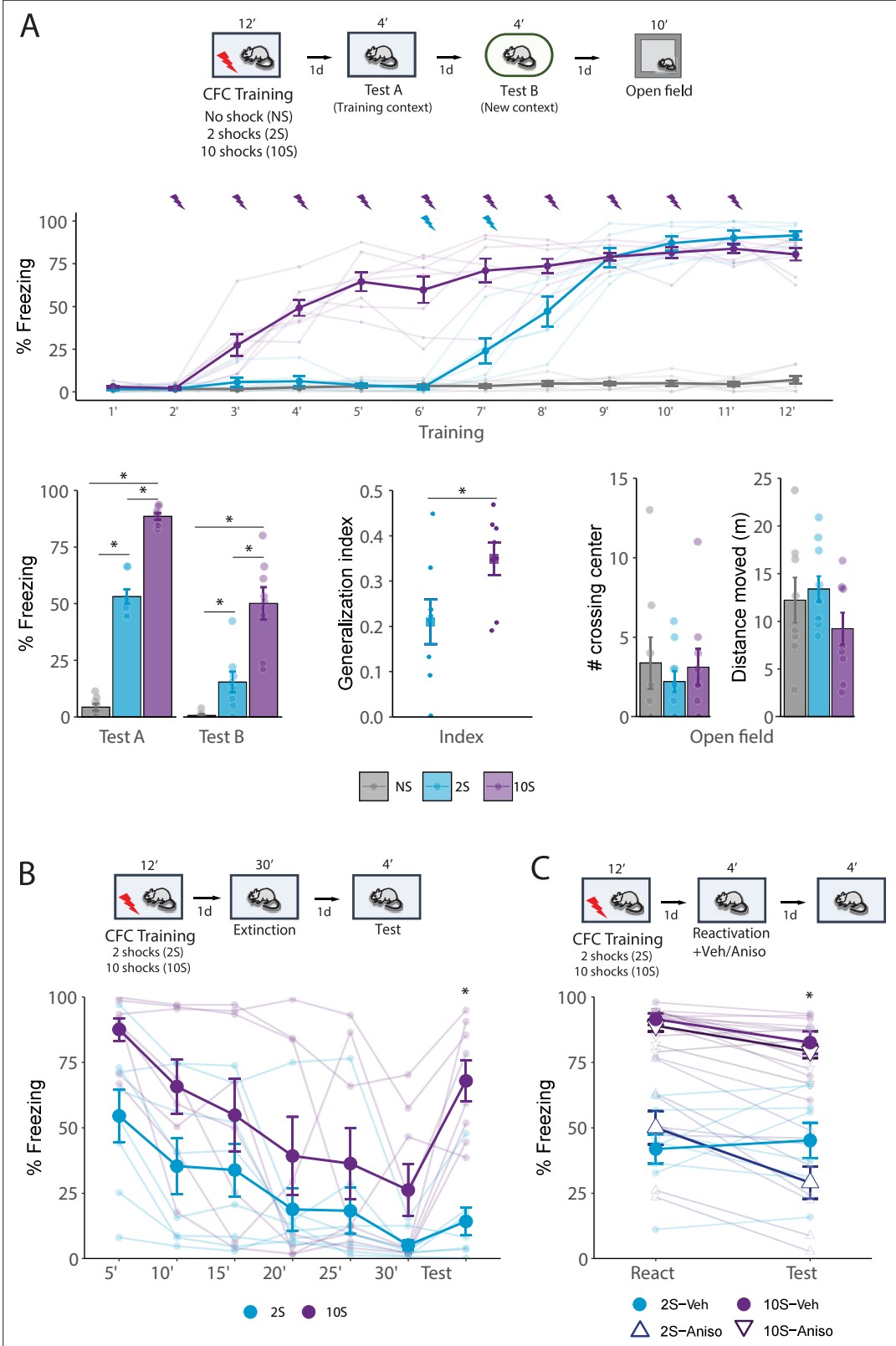

**Figure 1.** Behavioral differences triggered by contextual fear conditioning (CFC) using different numbers of foot shocks. (**A**) Animals were placed in a fear conditioning box for 10 min and received either no foot shock (NS group), 2 shocks (2S group), or 10 shocks (10S group). One day later, they returned to the same context for 4 min (Test A), and on the following day were placed for 4 min in a new, distinct context (Test B). The next day,

*Figure 1 continued on next page*

*Figure 1 continued*

rats were placed for 10 min in an open-field arena. During training, freezing behavior increased following shock presentations in both 2S and 10S groups (N = 6/8 per group). In both Tests A and B (left), animals trained with 10 shocks displayed higher freezing levels than other groups, and animals that received no shock displayed the lowest freezing. To assess how animals from the 2S and 10S groups differed regarding fear generalization, an index was calculated (Freezing in Test 2/Freezing in Test 1 + Test 2), revealing that 10S animals expressed higher generalization (center). In the open-field test, there were no differences between groups across parameters of exploratory behavior (right). (**B**) Animals were submitted to the 2S and 10S training protocols, and on the next day were re-exposed to the training context for 30 min to induce extinction learning. The next day, a 4 min test was performed to evaluate extinction retention. Throughout the extinction session, both groups exhibited a decrease in fear expression but 10S animals displayed significantly higher freezing. In the test session, 2S animals displayed extinction retention, with no increase in freezing in comparison to the last 5 min bin of the extinction session, whereas 10S animals displayed a return of fear to the levels of the first bin. (**C**) Animals were submitted to the 2S and 10S training protocols, and the next day the fear memory was reactivated by re-exposing animals to the training context for 4 min. Immediately after the reactivation session, anisomycin or its vehicle was injected i.p. to block memory reconsolidation. The efficiency of the reconsolidation was assessed the next day in a test where animals were again exposed to the conditioned context for 4 min. In the 2S group, anisomycin treatment led to a decrease in fear expression, whereas the same treatment was ineffective in the 10S group. Plots show the mean ± SEM. N = 8 per group. *p<0.05.

14.4, p<0.001), and no group × bin interaction ($F_{6,84}$ = 1.7, p=0.12). Tukey's post hoc revealed that the 2S group displayed extinction retention since there was no difference between the last extinction bin and the test session the next day (p=0.9), but in the 10S group there was a significant increase in freezing (p=0.003) that returned to the levels found in the first bin (p=0.72). Furthermore, in the 2S group, freezing during test remained consistent with the levels observed in the final extinction bin (p=0.61) and was lower than the levels in the initial extinction bin (p<0.001), indicating no spontaneous recovery. Conversely, in the 10S group, freezing levels increased from the final extinction bin to the test (p=0.003), reaching levels comparable to those observed in the first extinction bin (p=0.19), indicating complete spontaneous recovery.

Next, we investigated the effect of training intensity on the ability of memory to change through reconsolidation. One day after the initial training, memory was reactivated by a 5 min re-exposure to the training context, followed immediately by the injection of the protein synthesis inhibitor anisomycin. To verify if reconsolidation was disrupted by the treatment, animals were re-exposed to the context the next day (*Figure 1C*). Repeated-measures ANOVA indicated a significant effect between groups ($F_{1,28}$ = 97.9, p<0.001) and a group × session × treatment interaction ($F_{1,28}$ = 8.72, p=0.006). Tukey's post hoc revealed that in the 2S group, post-reactivation anisomycin led to a decrease in freezing during the test (p<0.001), while the freezing levels of vehicle-treated animals remained constant (p=0.9). In contrast, freezing in the 10S group did not change across sessions in both animals that received vehicle (p=0.36) and anisomycin (p=0.25). This suggests that memories formed with the stronger training protocol (10S group) are resistant to change upon recall.

These data demonstrate that memories formed with 2- and 10-shock protocols exhibit significant differences at the behavioral level. Secondly, they show that 10-shock training creates maladaptive memories that trigger intense and generalized fear responses, are resistant to suppression by extinction, and are unable to flexibly change through reconsolidation.

## Commonalities and differences in brain activation induced by recall of mild and strong fear memories

To investigate the neural mechanisms underlying the formation of mild and strong fear memories, we trained a new cohort of animals using the same CFC protocol as described in the previous experiments. We then performed immunohistochemistry (IHC) to count c-fos-expressing cells and measure brain activation during memory recall in 12 brain regions implicated with fear memory processes (*Figure 2*). Based on previous literature, we examined the following regions: the retrosplenial cortex (RSC), the infralimbic cortex (IL), the cingulate cortex area 1 and area 2 (Cg1 and Cg2), the dentate gyrus (DG) and the CA3 and CA1 hippocampal fields, the basolateral and central nuclei of the amygdala (BLA and CeA), and the paraventricular and the reuniens nuclei of the thalamus (PV and Re).

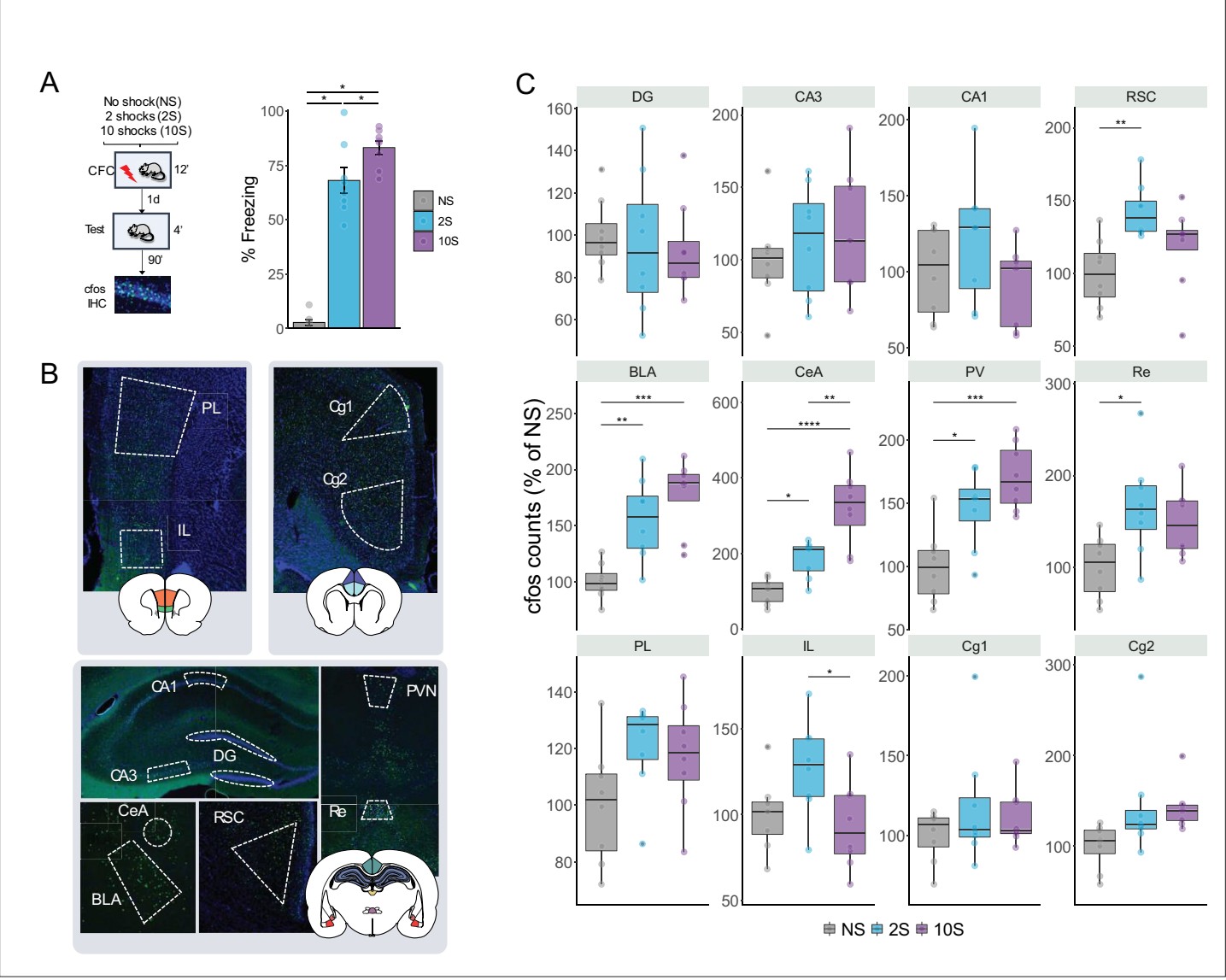

**Figure 2.** Retrieval-induced brain activation of neutral, mild, and strong fear memories. (**A**) Animals were submitted to the NS, 2S, or 10S training protocols. To trigger the recall of the resulting memory, 1 d after training rats returned to the same context for 4 min, and 90 min later, brains were extracted for c-fos immunohistochemistry. During the test, animals in the 10S displayed higher freezing levels than all others, and 2S animals froze more than NS animals. (**B**) Boundaries of the regions targeted for c-fos quantification. Twelve regions were analyzed: in the hippocampus, the dentate gyrus (DG) and the CA3 and CA1 subfields; the basolateral (BLA) and the central (CeA) nuclei of the amygdala; the retrosplenial (RSC), infralimbic (IL), prelimbic (PL), and cingulate cortices (Cg1 and Cg2); the reuniens (Re) and paraventricular (PV) nuclei of the thalamus. (**C**) In the BLA, CeA, and PV, both 2S and 10S displayed higher c-fos expression than NS. In the CeA and PV, both 10S also displayed higher c-fos expression than 2S. In the Re and RSC, only the 2S group showed higher c-fos counts than the NS. In the IL, c-fos expression was reduced in the 10S group in comparison to 2S. Plots show the mean ± SEM. N = 8 per group. *p<0.05; ** p<0.01; *** p<0.001; **** p<0.0001.

Similar to our first set of experiments, there was a significant difference in freezing expression between groups during test (one-way ANOVA: $F_{2,21}$ = 118.4, p<0.0001) and freezing expression increased according to the training intensity (Tukey's post hoc test: NS × 2S: p<0.0001; NS × 10S: p<0.0001; 2S × 10S: p=0.033).

In the c-fos analysis, the data were normalized to the average of the NS group (*Figure 2C*). ANOVA indicated a significant difference between groups in the following brain regions: RSC ($F_{2,21}$ = 6.39, p=0.007), Re ($F_{2,21}$ = 5.22, p=0.014), IL ($F_{2,21}$ = 4.18, p=0.03), BLA ($F_{2,21}$ = 14.9, p<0.0001), CeA ($F_{2,21}$ = 22.8, p<0.0001), and PV ($F_{2,21}$ = 12.9, p=0.0002). Tukey's post hoc test was then performed to reveal pairwise differences between groups within each structure. In the RSC and Re, c-fos levels were

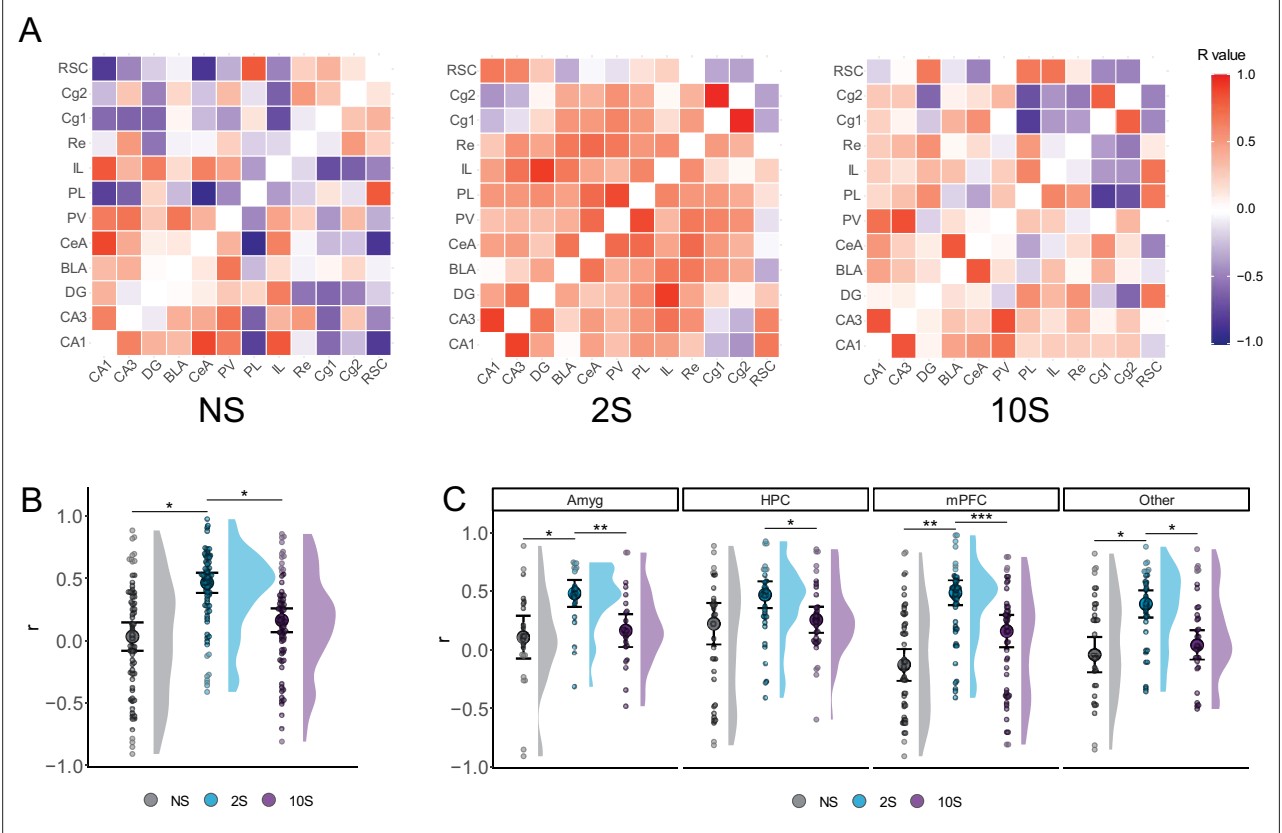

**Figure 3.** Altered functional connectivity during mild and strong fear memory recall. (**A**) Color-coded matrices showing the interregional correlation values in the NS, 2S, and 10S groups. The diagonals were zeroed. (**B**) Mean r values were higher in the 2S group than in the others. (**C**) The 2S group displayed higher mean r values than the other groups in all regions except in the HPC, where it did not differ from 10S. Plots show the median ± CI. *p<0.05; ** p<0.01; *** p<0.001.

higher than NS controls only in the 2S group (p=0.005 and p=0.01, respectively), while in the IL c-fos was reduced in the 10S group in comparison with the 2S group (p=0.03). Both 2S and 10S groups displayed higher c-fos levels than NS controls in the BLA (NS × 2S: p=0.003; NS × 10S: p<0.0001), CeA (NS × 2S: p=0.042; NS × 10 S: p=0.001), and PV (NS × 2 S: p=0.010 NS × 10S: p=0.0001). In addition, there was an increase in c-fos in the 10S group in comparison with the 2S group (p=0.0014). These results show that the retrieval of mild and strong fear memories triggers different patterns of brain activation across different brain regions.

## Coordinated interregional brain activity is decreased during strong fear memory recall

When the activity of a pair of brain regions varies together, it can be inferred that these regions are functionally coupled (*Wheeler et al., 2013*). To further understand the mechanisms underlying the recall of memories with varying levels of aversiveness, we investigated if there were changes in the coordinated activity between brain regions. Therefore, we assessed the synchrony in neuronal activation across brain regions by computing the Pearson correlations between all pairs of brain regions within each group (*Figure 3A*).

We first compare the distribution of r values between groups (*Figure 3B*) using Kruskal–Wallis followed by Mann–Whitney tests. This revealed a significant difference between groups regarding r values (H = 0.118, df = 2, p<0.001), such that the r values of the 2S group were larger than those of the NS (p<0.001) and 10S (p<0.001) groups, which did not differ from one another (p=0.13).

Next, we conducted the same analysis but compared the r values in collections of brain regions (*Figure 3C*): the amygdala (Amyg: CeA and BLA) hippocampal fields (HPC: DG, CA3, and CA1), the prefrontal cortex (PFC: PL, IL, Cg1, and Cg2), and the remaining structures that were quantified

(Other: RSC, PV, and Re). There was a significant difference between groups in the amygdala (H = 0.157, df = 2, p=0.003), hippocampus (H = 0.05, df = 2, p=0.03), prefrontal cortex (H = 0.166, df = 2, p<0.001), and in the remaining structures (H = 0.08, df = 2, p=0.009). Subsequent Mann–Whitney tests revealed that, in the 2S group, the r values were significantly larger than the NS or 10S groups in all structures (p<0.05) except in the hippocampus, where it did not differ from 10S rats (p=0.14). In all regions, NS and 10S groups did not differ (p>0.05).

The results indicate that the level of coordinated activity between brain regions, as indicated by r values, changes during the recall of memories with varying levels of aversiveness. Compared to the re-exposure to a neutral context (NS group), the recall of a mild fear memory (2S group) was found to be associated with an increase in the level of coordinated activity across brain regions. However, this increase in coordinated activity was not observed during the recall of a severe fear memory (10S group).

## Network-level changes in the brain underlie the intensity of fear memories

Having established that the levels of coordinated activity during recall vary according to the intensity of the memory, we employed graph theory to construct functional networks and gain a deeper understanding of the interactions between brain systems in each condition.

The networks were constructed such that each node represents a brain region, and edges connecting the nodes are weighted according to the respective correlation coefficients (*Figure 4A*). To generate functional networks, thresholding was applied to retain only the edges with the highest weights, which indicate the strongest coordinated activity and the most essential functional connections. This was accomplished by excluding edges with r values lower than the average plus 1 SD of all networks (*Figure 4B*; r < 0.61).

To measure the relevance of individual nodes within the networks and gather insight into how the information flows within the networks, the degree centrality, betweenness centrality, and nodal efficiency were computed. The nodes were ranked according to each metric to reveal their relative importance in the respective network. The degree centrality measures the number of connections each node in the network has to other nodes (*Figure 4C*). The betweenness centrality measures the instances in which a node acts as a bridge along the shortest path between two other nodes (*Figure 4D*). The nodal efficiency measures the inverse of the average shortest path length between a node and all other nodes in the network and reflects its ability to exchange information with other nodes, whereas the global efficiency reflects how well the network communicates as a whole and is computed as the average nodal efficiency (*Figure 4E*).

In the 2S network, the CeA and the BLA were among the highest-ranked nodes across measurements, indicating a central and well-connected position in the network. In contrast, the amygdala nuclei ranked low in the other groups. In the NS group, this is consistent with the lack of conditioned fear observed in these animals. In the 10S group, however, where conditioned fear was highly expressed during recall, this lower ranking suggests a reduction in the functional interactions between the amygdala and the other brain regions. Therefore, the brain-wide functional integration of the CeA and BLA observed in the mild fear memory network seems to be reduced in the severe fear memory network.

To gain a better understanding of how each network differed regarding their connectivity patterns, centrality measures were compared using Kruskal–Wallis followed by Mann–Whitney tests. When analyzing degree (*Figure 4C*, right), it was found that there was a significant difference between groups (H = 0.226, df = 2, p=0.009), with the 2S network displaying higher degree values than NS and 10S (p<0.05), which did not differ from each other (p=0.81). Regarding betweenness (*Figure 4D*, right), there was a significant overall group difference (H = 0.272, df = 2, p=0.005) and the mild memory network scored higher than the strong (p=0.009) but not the NS network (p=0.06), and the NS and 10S networks did not differ (p=0.25). The values of global efficiency also differed among the memory networks (*Figure 4E*, right; H = 0.605, df = 2, p<0.001). The 2S network displayed higher values than NS and 10S networks (p<0.001), which did not differ from each other (p=0.32).

A feature of many complex networks, including anatomical and functional brain networks, is a small-world organization, which is characterized by a balance of high local clustering but a small number of steps separating each node (*Bullmore and Sporns, 2009*). This balance enables rapid

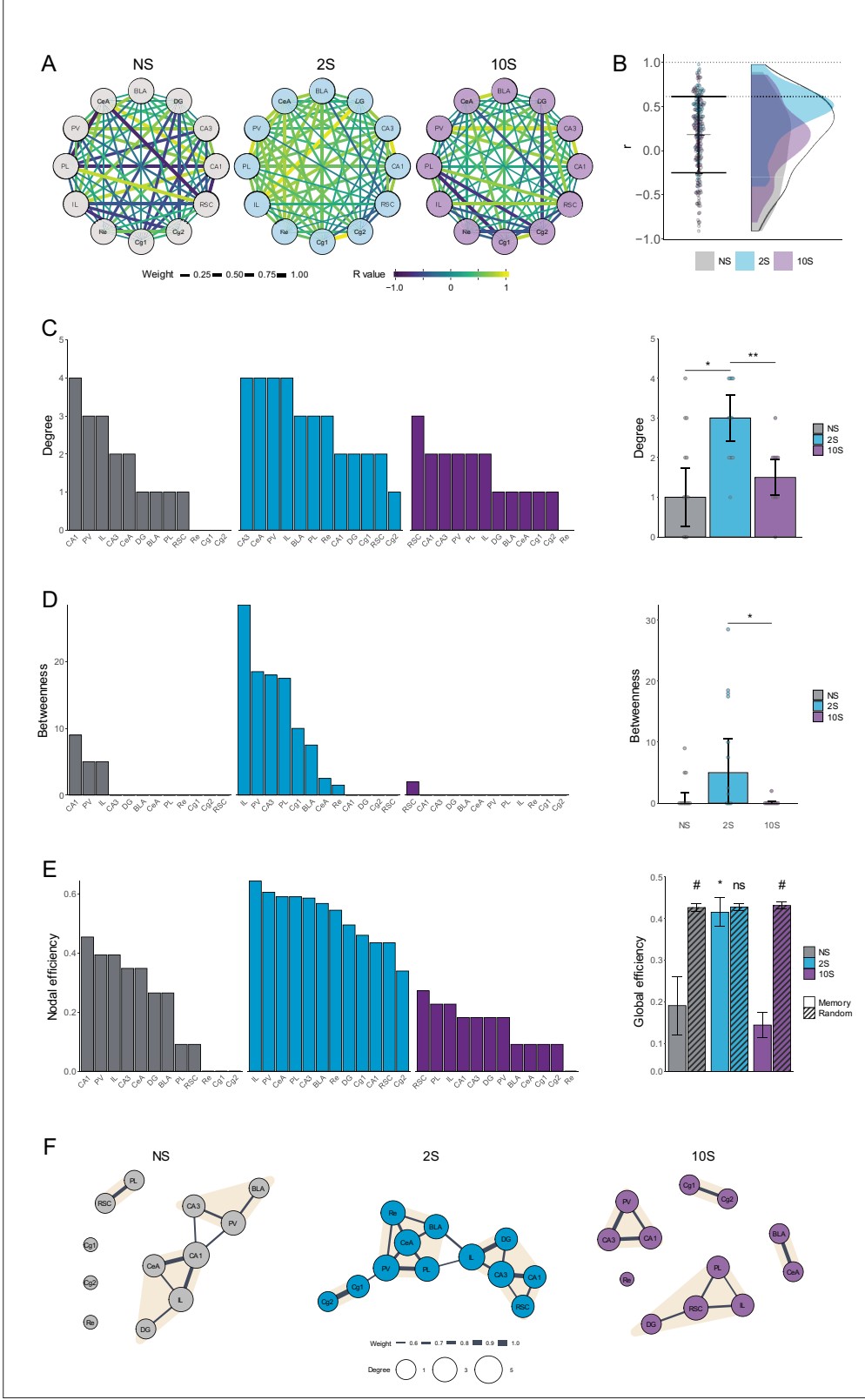

**Figure 4.** Analysis of functional brain networks engaged by the recall of mild and strong fear memories. (**A**) Unfiltered networks comprising nodes representing brain regions, and edges representing the intercorrelation value between each pair of regions. The edges' width scale with its weight (absolute r-value) and colors indicate the real r-value. (**B**) R values that were retained after thresholding (between dotted lines) and assigned as edges

*Figure 4 continued on next page*

*Figure 4 continued*

in the functional memory networks. (**C**) Left: degree distribution in each group. Right: the mean degree was higher in the 2S group than in NS and 10S groups. * p<0.05; ** p<0.01. (**D**) Left: betweenness distribution in each group. Right: the mean betweenness was higher in the 2S than in NS and 10S. * p<0.05. (**E**) Left: nodal efficiency distribution in each group. Right: the global efficiency was higher in the 2S than in NS and 10S, and only in the 2S group it matched that of random networks with small-world configuration. * p<0.05. # Confidence interval for the difference between means > 95%.(**F**) Representation of the functional networks of the NS (left), 2S (center), and 10S (right) groups. The nodes represent brain regions and the edges represent the functional connections between them. The nodes' sizes are scaled according to their degree, and the edges are scaled according to their r value. Nodes detected as being part of the same cluster are highlighted with the same color. Plots show the median ± CI.

specialized local processing and efficient global communication, resulting in a robust and efficient network structure. To determine whether the memory networks generated in this study display global efficiency comparable to that of small-world networks, random networks were generated using the Watts–Strogatz small-world model (*Watts and Strogatz, 1998*). The global efficiencies of the memory and random small-world networks were then compared through the calculation of the confidence interval for the difference between means (*Figure 4E*, right). The 2S memory network matched the global efficiency of its respective random networks (mean difference = 0.012, 95% CI [–0.031, 0.056]). In contrast, the global efficiency of the memory networks was lower than those of the respective random networks in the case of the NS group (mean difference = 0.236, 95% CI [0.161, 0.309]) and of the 10S group (mean difference = 0.288, 95% CI [0.284, 0.327]).

These results demonstrate that, among the brain structures measured here, the level of aversive intensity of memory has a critical impact on functional connectivity during recall. In comparison to the neutral and severe fear memory networks, the mild fear memory network displayed higher functional connections between brain regions (degree), more nodes connecting different parts of the network (betweenness), and higher efficiency in global information flow that was consistent with that of small-world networks (global efficiency). Overall, this suggests that during mild fear memory recall, fear expression driven by the amygdala is modulated by a large, integrated interregional system, but in strong fear memories this modulation is disrupted.

This visualization of functional networks shows a closely connected network for the mild memory group, in contrast to the fragmented networks observed for the neutral and strong fear memory groups (*Figure 4F*). Clusters in the networks were identified using a leading eigenvector method (*Newman, 2006*). It revealed that in the mild memory network, there is a central cluster composed of the amygdala, Re, PV, and PL. A second cluster included the hippocampus, the IL, and RSC, and a third cluster comprised the cingulate cortex. This suggests that the amygdala is well positioned for interregional communication, potentially allowing for modulatory control of fear expression. This is not the case in the strong memory network where the BLA and CeA are close to each other but isolated from other regions, resulting in less modulatory control of fear.

## Discussion

This study investigated how functional connectivity during fear memory recall varies depending on fear memory intensity. Utilizing mild and strong fear protocols, we were able to generate memories that exhibit remarkable differences in their behavioral and neuronal signatures. Specifically, we found that mild memories triggered moderate and precise behavioral responses that were susceptible to extinction and reconsolidation, while strong memories triggered robust and generalized responses and were resistant to both extinction and reconsolidation. When comparing the brain activity across 12 regions implicated in fear memory, we found that recall resulted in changes in overall activity in only a subset of these regions. However, when we examined coordinated activity patterns, we discovered striking differences in the functional networks that underlie mild and strong fear memory recall. Mild memories displayed a well-connected network with the amygdala, which was well-positioned to interact with and be influenced by several regions. However, such connectivity was absent in the strong fear memory network, which may underlie the observed high and generalized fear responses that are resistant to attenuation. These findings suggest that abnormal functional connectivity may be a core factor in maladaptive, severe fear memories.

To study brain differences between mild and strong fear memories, we used conditioning protocols of different intensities that generated fear memories remarkably differently at the behavioral level. Conditioning with 2 shocks resulted in a fear memory eliciting mild and discriminative fear responses, which could be attenuated by extinction, and its recall triggered reconsolidation. In contrast, a 10-foot shock training generated a strong fear memory that can be seen as maladaptive; it elicited high and generalized fear expression, which was resistant to undergo both extinction and reconsolidation. Importantly, our control animals were exposed to the conditioning chamber for an equivalent duration without being subjected to shocks, thus encoding and recalling a nonfearful contextual memory (*Figure 1*). The recall of these distinct memories resulted in differential brain activation in some structures. In line with the role of the amygdala (*Bernabo et al., 2021*; *Gründemann et al., 2019*; *Josselyn et al., 2015*; *Maddox et al., 2019*) and paraventricular nucleus (*Do-Monte et al., 2015b*; *Padilla-Coreano et al., 2012*) in fear expression, we observed that c-fos counts in these structures increased as a function of the strength of the fear memory. Conversely, in the infralimbic cortex, c-fos expression was decreased in the 10S group, which is in line with its established role in safety learning and fear suppression (*Bloodgood et al., 2018*; *Do-Monte et al., 2015a*). Adding to the known involvement of the RSC (*Todd et al., 2019*; *Trask et al., 2021*) and the Re (*Ramanathan et al., 2018*; *Sierra et al., 2017*; *Troyner et al., 2018*) in modulating fear memory formation and expression, our results revealed neural activity in these structures was increased during mild memory recall, but not during strong memory recall, when compared to unshocked controls. Although the hippocampus (*Gründemann et al., 2019*; *Josselyn et al., 2015*; *Guo et al., 2018*; *Haubrich and Nader, 2018b*; *Pedraza et al., 2016*), prelimbic (*da Silva et al., 2020*; *Dixsaut and Gräff, 2022*; *Fernandez-Leon et al., 2021*), and cingulate cortex (*Finnie et al., 2018*; *Haubrich et al., 2016*; *Ortiz et al., 2019*) are critical to contextual fear memory-related processes, we found no differences in their activation between groups, suggesting that the role of these structures in modulating the expression of fear of different intensities may not reflect in changes in overall activation levels.

The amygdala is a central hub for the processing of fear memory and fear expression during recall. It receives input from sensory areas and integrates this information with several other brain regions, such as the hippocampus and the prefrontal cortex, to generate adaptive fear responses. The projections to the amygdala from these regions modulate the intensity of fear responses, extinction learning, and the induction of reconsolidation. Consequently, the functional connectivity of the amygdala within a memory network plays a crucial role in determining the level of fear expression. Our initial assessment of mean coordinated activity supports this concept. We found that, in comparison with unconditioned controls, the overall correlation values of the amygdala with other brain regions increase during the recall of a mild fear memory, but not during the recall of a strong fear memory.

The functional networks analysis further evaluated the system-level interactions of different brain regions during the recall of mild and severe fear memories. Centrality measures were computed to assess the number of functional connections, hub nodes, and information flow throughout the network. These measures were increased during the recall of mild fear memories compared to untrained controls, and such an increase was not observed during the recall of strong fear memories. Moreover, only the mild fear memory network displayed a small-world organization, which allows for both specialized and efficient information processing in the network. These findings suggest that the expression of moderate and flexible fear memories involves an efficient, integrated activation of specific brain regions, which is not present in the recall of strong fear memories.

Importantly, both the basolateral and the central amygdala ranked high across the centrality measures assessed. Of particular importance is the high nodal efficiency ranking of the amygdala, indicating that it was a main central mediator of information flow throughout the network. Notably, the infralimbic cortex, a region known to suppress fear responses by sending inhibitory projections to the amygdala (*Bouton et al., 2021*), displayed the highest betweenness centrality during mild memory recall. However, during the recall of severe fear memory, the amygdala displayed a considerably lower ranking across centrality measures, similar to what was observed in control unconditioned animals. These changes in the network-level integration of the amygdala are apparent in the model depicted in *Figure 4F*. In the 2S graph, the amygdala is at the center of a fully connected network and the IL is a critical bridge connecting the amygdalar module to the one containing the hippocampus and the retrosplenial cortex. In contrast, in the 10S graph, the amygdala is disconnected from the other regions. This indicates that the differences observed behaviorally during severe fear memory

recall – high and generalized freezing and resistance to attenuation – coincided with a disruption in the functional connectivity of the amygdala with other regions that are important for regulating fear. To further elucidate the underlying mechanisms of fear memory strength in vivo, understanding the specific roles of individual network elements in fear regulation becomes essential. Future research will be important to probe the causal interplay among distinct nodes and edges, both individually and in combination, in shaping diverse aspects of fear expression.

To generate mild and strong fear memories, we based our conditioning parameters on methods that have shown distinct behavioral outcomes in prior studies (*Holehonnur et al., 2016*; *Wang et al., 2009*; *Poulos et al., 2016*). To ensure a focused comparative analysis, our conditioning protocols differed only in the number of foot shocks and maintained consistent shock intensities and session durations. Yet, the number of shocks is not the only factor that can affect the strength of fear memories (*Gazarini et al., 2023*). Other conditioning parameters, such as shock intensity, its predictability, and inter-shock intervals, can also play crucial roles. Moreover, different fear measures like freezing behavior, fear-potentiated startle, and inhibitory avoidance might manifest differently following varying conditioning protocols, adding another layer of complexity. A comprehensive understanding of fear memory strength will benefit from further studies scrutinizing these parameters and memory attributes. In addition, a growing body of evidence underscores the differences between males and females concerning fear memories (*Fleischer and Frick, 2023*). Given that our study was conducted only with male rats, future studies exploring sex differences will be instrumental in providing a more complete account of the network-level mechanisms underlying fear memory strength.

In summary, this study investigated the behavioral and neural differences during the recall of memories with varying levels of aversiveness. In contrast to memories formed with a 2-shock fear conditioning protocol, 10 shocks generate maladaptive memories with intense and generalized fear responses, resistance to extinction, and an inability to flexibly change through reconsolidation. The recall of these memories resulted in changes in brain activation across some, but not all, regions where c-fos was measured. Graph-based network analysis uncovered that moderate and flexible fear memory expression involves an efficient integrated activation of particular brain regions, which is absent in strong fear memories. These results provide a deeper understanding of fear memory by showing that functional connectivity during recall varies greatly depending on the level of aversiveness of the memory. It supports the notion that not only local synaptic mechanisms but also broader functional changes play a role in shaping fear memory strength. Moreover, these findings highlight the importance of studying the neural mechanisms involved with fear memory strength, as they may provide insight into the development of maladaptive memories and potential interventions for their treatment.

## Materials and methods

### Subjects

Male Sprague–Dawley rats (275–300 g at arrival; Charles River, Quebec, Canada) were housed in pairs in plastic cages in a temperature-controlled environment (21–23°C) with ad libitum access to food and water and maintained on a 12 hr light/dark cycle (lights on at 7:00 A.M.). All experiments were conducted during the light phase. In all experiments, animals were randomly assigned to each behavioral condition. Sample size estimates were determined based on effect sizes observed in previous reports using similar assays (*Wang et al., 2009*; *Haubrich et al., 2020b*), resulting in statistical power estimates between 70 and 90%. Each rat was handled for at least 5 d before the behavioral procedures. All procedures were approved by McGill's Animal Care Committee (Animal Use Protocol #2000-4512) and complied with the Canadian Council on Animal Care guidelines.

### Drugs

Anisomycin (150 mg/kg, i.p.; Sigma-Aldrich) was dissolved in equimolar HCl and sterile saline and had its pH adjusted to 7.5.

### Contextual fear conditioning and behavioral testing

The training session, which lasted 12 min in all groups, took place in transparent Plexiglas conditioning boxes enclosed in soundproofed chambers (30 × 25 × 30 cm; Coulbourn Instruments) with a fan on as

background noise. Animals assigned to the 2-shocks group (2S) received 2 × 1 s × 1.0 mA foot shocks at the sixth and seventh minutes, while animals in the 10-shocks group (10S) received foot shocks every minute from the second to the eleventh minute. A stainless-steel grid floor provided the shocks.

The recall and extinction sessions were conducted in the same context as the fear conditioning session, with recall lasting 4 min and extinction lasting 30 min. The generalization test, which also lasted 4 min, was conducted in different conditioning boxes with white-striped front and back walls and rounded white plastic walls on the sides, as well as a white plastic floor. Different sessions were always conducted 24 hr apart. Behavior was recorded by digital cameras and memory was evaluated by a blind experimenter measuring the time spent freezing, using Freeze View software (Actimetrics). In the analysis of fear expression during the training session, due to technical issues, we could only assess six out of eight animals in the 2S group. Freezing was defined as immobilization except for respiration. The generalization index was calculated as Freezing in Test B/(Freezing in Test A + Freezing in Test B).

The open-field test lasted 5 min and was conducted in a 1 m$^2$ gray Pexiglass arena. Behavior was recorded by digital cameras and analyzed with the software EthoVision (Noldus, The Netherlands).

## Immunohistochemistry

After 90 min of testing, rats were anesthetized and perfused with buffered saline followed by 4% paraformaldehyde (PFA). The brains were then collected, fixed in 4% PFA overnight, transferred to a 30% sucrose solution, and stored at 4°C until cryo-sectioning at 50 μm thickness. To measure c-fos expression, coronal brain slices were incubated in a blocking solution at room temperature for 1 hr (3% NGS, 0.3% Triton X-100) and then for 20 hr with anti-c-fos primary rabbit antibody (1:500, 226.003; Synaptic Systems, Göttingen, Germany). Sections were washed and incubated with anti-rabbit Alexa-488 secondary antibody (1:500, Jackson Immunoresearch, West Grove, PA) for 2 hr at room temperature. The sections were washed again and mounted on slides and immediately coverslipped with Fluoromount-G with DAPI (Thermo Fisher). Images were examined under fluorescence microscopy (Leica DM 5000 B) and c-fos-positive cells were counted bilaterally from at least two slices for each animal with ImageJ.

## Statistics and networks generation

To analyze the results of the behavioral tests and c-fos expression, we used two-tailed independent-samples t-test, one-way, two-way, or two-way repeated-measures ANOVA for data analysis. Tukey's post hoc tests were further used to identify the specific differences that contributed to significant interactions. Type 1 error rate was set at 0.05.

For the analysis of functional connectivity, Pearson correlation coefficients were computed for all pairwise correlations of c-fos expression levels across the 12 targeted regions of interest within each group. To compare mean r values between groups, we used the Kruskal–Wallis test followed by post hoc Mann–Whitney tests.

The functional networks were generated by considering only the strongest correlations, as determined by a threshold of Pearson's r values that were higher than the mean plus the SD of all correlations. This thresholding approach was used to provide a cut-off based on the data's inherent distribution, thereby retaining the top edges according to the data variance. The networks were generated and centrality measures were computed using the R package igraph (igraph.org/r) and custom R code (https://github.com/johaubrich/Networks; copy archived at *Haubrich, 2023*). The nodes in the network represent brain regions, and the correlations retained after thresholding were considered as edges, with the r value being assigned as the edge's weight. Degree denotes the total number of edges connected to a particular node. Betweenness represents the number of times a node appears on the shortest paths between two other nodes. The shortest path is the length of the path connecting two nodes with the lowest number of edges. The nodal efficiency of each node was calculated as the average inverse of the shortest path length to all other nodes in the network. For the nodal efficiency calculation, the average path distance considered the weight of the edges (i.e., 1/r value). The global efficiency denoted the average nodal efficiency of all nodes within a network. Comparisons of degree, betweenness, and global efficiency were conducted using Kruskal–Wallis tests followed by Mann–Whitney tests. For the small-world analysis, we generated 1000 random networks for each group using the Watts–Strogatz small-world model (*Watts*

*and Strogatz, 1998*). To ensure that the random networks had a similar density and could provide a suitable comparison for global efficiency, these networks were generated enforcing a similar clustering coefficient (*Barrat et al., 2004*) to their respective memory network (calculated with 1000 iterations per network). The global efficiency of the networks for the NS, 2S, and 10S groups was compared to their respective random networks by calculating the 95% confidence interval of the difference between means.

All statistical analysis and plots were generated using RStudio (R version 4.2.2, RStudio version 2022.12.0).

## Acknowledgements

We are grateful to Karine Gamache for technical support and Karine Gamache, Matteo Bernabo, and Isabelle Groves for their insights during the writing of the manuscript. This work was supported by the Natural Sciences and Engineering Research Council of Canada (203523) and Canadian Institutes of Health Research (238757).

## Additional information

### Funding

| Funder | Grant reference number | Author |
|---|---|---|
| Natural Sciences and Engineering Research Council of Canada | 203523 | Josue Haubrich Karim Nader |
| Canadian Institutes of Health Research | 238757 | Josue Haubrich Karim Nader |

The funders had no role in study design, data collection and interpretation, or the decision to submit the work for publication.

### Author contributions

Josue Haubrich, Conceptualization, Data curation, Formal analysis, Validation, Investigation, Visualization, Methodology, Writing – original draft, Project administration, Writing – review and editing; Karim Nader, Conceptualization, Resources, Supervision, Funding acquisition, Project administration, Writing – review and editing

### Author ORCIDs

Josue Haubrich ⓘ https://orcid.org/0000-0002-3632-5566

### Ethics

All procedures were approved by McGill's Animal Care Committee (Animal Use Protocol #2000-4512) and complied with the Canadian Council on Animal Care guidelines.

Reviewer #1 (Public Review): https://doi.org/10.7554/eLife.88172.3.sa1
Reviewer #2 (Public Review): https://doi.org/10.7554/eLife.88172.3.sa2
Author Response https://doi.org/10.7554/eLife.88172.3.sa3

## Additional files

### Supplementary files
• MDAR checklist

### Data availability
All data generated and analyzed during this study are available at https://doi.org/10.5061/dryad.280gb5mw3.

The following dataset was generated:

| Author(s) | Year | Dataset title | Dataset URL | Database and Identifier |
|---|---|---|---|---|
| Haubrich J, Nader K | 2023 | Network-level changes in the brain underlie fear memory strength | http://doi.org/10.5061/dryad.280gb5mw3 | Dryad Digital Repository, 10.5061/dryad.280gb5mw3 |

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
