## [Editor Report · eLife assessment]

This **important** study provides **convincing** data in support of the conclusion that weak but not strong fear memories are more easily modified using behavioral and pharmacological approaches potentially as a result of differential connectivity with the amygdala showing greater connectivity through the brain in weak compared to strong memories. The scope of the paper would be strengthened if both sexes were examined and more varied definitions of weak vs. strong memories were used. This paper is of interest to behavioral and neuroscience researchers studying learning, memory, and/or neural networks.

---

## [Referee Report · Reviewer #1 (Public Review)]

The authors demonstrate that reactivation of mild vs strong aversive contextual associations produces dissociable effects on fos expression across a wide network of relevant brain regions. Mild, 2-shock memory recruits a 'small-world' network in which amygdalar regions are functionally connected to other regions that modulate their activity and behavioral output, whereas strong, 10-shock memory isolates amygdalar nuclei from the rest of the network. These different patterns of correlated neural activity correspond with functional/behavioral differences - the authors confirm that weak, 2-shock memory is more effectively extinguished and is susceptible to reconsolidation relative to strong, 10-shock memory.

One major drawback of the manuscript is the fact that the data were collected from male subjects only. One might expect similar behavioral outcomes from male and female rats receiving 2-shock and 10-shock training. However, increasing attention to sex as a biological variable has revealed an interesting truth, namely that males and females can engage distinct neural pathways to arrive at the same behavioral destination. It should not be taken for granted that retrieval of aversive contextual associations would reproduce the same networks in females, and, as such, the manuscript does not give a complete accounting of the phenomenon under study.

---

## [Referee Report · Reviewer #2 (Public Review)]

The manuscript examined the behavioural and neural profile of weak and strong fear memories. The data provide strong evidence that weak but not strong fear memories are subject to extinction and reconsolidation disruption. Strong memories also show greater generalization. These differences were echoed in differential neural connectivity with weak fear memories showing greater connectivity between brains areas than strong fear memories.

The findings are of a great importance and offer insight into why resistance to extinction and reconsolidation may underlie fear-related psychopathology.

The study uses key behavioural tests to study the durability of weak vs strong memories (extinction and reconsolidation) as well as studies the generalisation of those memories. These behavioural effects nicely dovetail with the neural connectivity analyses that were performed.

The data presented in this paper will be the basis for future hypothesis driven examinations on the causal influence of specific pathways involved in contextual fear.

Excellent use of the open field to control for motor effects.

This is a strong paper and the results support the conclusions. The findings are of broad interest and are important for future research.

---

## [Author Response]

The following is the authors’ response to the original reviews.

We would like to thank you for your thoughtful review and constructive feedback on our manuscript. We have implemented numerous revisions throughout the manuscript to address your comments and suggestions. Below, our point-by-point responses to the reviewers' remarks. We hope that our revisions adequately address all raised concerns.

**Reviewer #1**
One major drawback of the manuscript is the fact that the data were collected from male subjects only. One might expect similar behavioral outcomes from male and female rats receiving 2shock and 10-shock training. However, increasing attention to sex as a biological variable has revealed an interesting truth, namely that males and females can engage distinct neural pathways to arrive at the same behavioral destination. It should not be taken for granted that retrieval of aversive contextual associations would reproduce the same networks in females, and, as such, the manuscript does not give a complete accounting of the phenomenon under study.

We thank the reviewer for highlighting the importance of sex differences in fear memory and for encouraging us to discuss this issue. We agree that males and females can engage different behavioral and circuit mechanisms and that our findings may not be generalizable to female rats. We expanded the discussion section to state this limitation and to suggest future directions for research on sex differences in fear memory:

“In addition, a growing body of evidence underscores the differences between males and females concerning fear memories (Fleischer and Frick, 2023). Given that our study was conducted only with male rats, future studies exploring sex differences will be instrumental in providing a more complete account of the network-level mechanisms underlying fear memory strength.”

The aversive associative memories described by the authors are characterized as mild or strong. More analysis of the meaning of memory strength, and its relationship to conditioning parameters, is needed.In particular, the authors should discuss issues such as amount of training, US magnitude, and rate of shock delivery. If amount of training is important, would 2 vs 10 presentations of a milder shock produce the same networks at retrieval? Would a larger shock require fewer presentations to isolate amygdalar regions from the rest of the network? If the shocks were presented at the same rate during training, would you get the same result in both groups? More data examining these questions would be ideal, but, in the absence of that, a discussion of these issues is needed and missing from the manuscript in its current form.

We appreciate the reviewer's feedback on the characterization of the fear memories in our study and agree that the labels "mild" and "strong" could oversimplify the complex nature of fear memories. Our study's main objective was not to delineate how varying conditioning protocols result in 'mild' or 'strong' fear memories, but to employ protocols of different intensities known to produce distinct behaviors, and then discern their brain differences. Our categorization was rooted in the resulting behavioral expressions, classifying 'mild' memories as those triggering sub-maximal fear responses with low generalization and a potential for extinction learning and reconsolidation. Conversely, 'strong' memories were defined by peak or near-peak fear responses, high generalization, and impeded extinction and reconsolidation processes. To isolate the number of foot shocks as the sole variable, we kept both shock intensity and session duration constant. While this decision allowed for a clear comparative analysis, we acknowledge its limitations in exploring other influential factors.

A more ideal approach would be to reverse this process—first experimenting with several different conditioning parameters and then observing the resulting behaviors and brain mechanisms—but given the additional workload that would entail, particularly when combined with the c-fos and network analyses, we opted for our current approach. Nevertheless, we hope our study will stimulate research that goes deeper into the nuances of fear conditioning protocols, fostering a better understanding of adaptive and maladaptive fear memories. This is now discussed in the discussion session:

“To generate mild and strong fear memories, we based our conditioning parameters on methods that have shown distinct behavioral outcomes in prior studies (Haubrich et al., 2020, 2015; Holehonnur et al., 2016; Poulos et al., 2016; Wang et al., 2009). To ensure a focused comparative analysis, our conditioning protocols differed only in the number of foot shocks, and maintained consistent shock intensities and session durations. Yet, the number of shocks is not the only factors that can affect the strength of fear memories (Gazarini et al., 2023). Other conditioning parameters, such as shock intensity, its predictability, and inter-shock intervals, can also play crucial roles. Moreover, different fear measures like freezing behavior, fear-potentiated startle, and inhibitory avoidance might manifest differently following varying conditioning protocols, adding another layer of complexity. A comprehensive understanding of fear memory strength will benefit from further studies scrutinizing these parameters and memory attributes.”

**Reviewer #2**
One alternative account to the weak vs. strong memory distinction made in the paper is the opportunity for extinction in the 2S compared to the 10S group. In the 2S group, the last shock occurs in the 3rd minute, leaving 9 minutes of context exposure without reinforcement to follow. This is not the case for the 10S group. If context fear extinction is engaged during this time, then this would mean that two memories (acquisition and extinction) are taking place in the 2S group, weakening the fear memory in this group, setting up the ground for stronger effects of extinction, less generalization and of course potential greater connectivity required for representing and linking these memories. Indeed, the IL, a brain area linked to extinction, is more predominant in the connectivity map of the 2S compared to the 10S group. While testing this alternative is beyond the scope of this paper, it will need to be discussed.

We thank the reviewer for raising this interesting point. We agree that the structure of the 2S protocol might inadvertently provide an opportunity for within-session extinction. However, we would like to clarify that we made a mistake in the description of the 2S training protocol. The timing of the shock deliveries was not at the second and third minutes as stated (a usual protocol in the literature), but at the sixth and seventh minutes. We apologize for this mistake and are thankful for your help in identifying this discrepancy which had unfortunately persisted despite multiple proofreading rounds. We have amended this detail in the methods section of our manuscript.

Nevertheless, we recognize that the subsequent minutes post-shock in the 2S group may still provide a window for potential extinction. To address this possibility, we scored the freezing expression during the training session minute by minute. In the 2S group, two videos were corrupted, and it was only possible to score freezing in six out of eight animals (this is acknowledged in the methods section). As presented in Figure 1.A (middle plot), freezing behavior increased post-shocks and showed no decline towards the session's end. These findings suggest that within-session extinction did not occur during our conditioning session. This analysis is now integrated into the relevant results subsection.

Methodological detail is lacking re the timeline of study, and connectivity analyses.

Thank you for your feedback. The formula for the discrimination index is now explained in the methods section. The new plot showing freezing behavior during training shows the exact time bin when shocks were delivered. We have expanded the description of the connectivity analysis.

**Reviewer #3 (Major concerns)**
1. Previous studies including Karim's lab have shown that protein synthesis in the hippocampus is required for the reconsolidation of contextual fear memory and that the retrieval of contextual fear memory activates gene expression such as c-fos in the hippocampus. However, the authors failed to confirm this observation. This may be due to the small number of rats or some technical problems.

Thank you for this insightful observation. We believe that the absence of the expected increase in hippocampal c-fos activation is due to the unique experimental design employed for our control group. In our study, control rats were subjected to an equivalent duration of context exposure without receiving shocks. As a result, these animals formed and retrieved a neutral, rather than fearful, contextual memory. This likely elevated cfos levels in the hippocampus in comparison to the more traditional home-cage condition frequently used in earlier studies. We used the NS (no shock) protocol for our control group to specifically elucidate the impact of the number of shock presentations on memory formation, therefore the context exposure was kept the same across groups. Importantly, this aspect did not affect our connectivity analysis, since it is influenced by the relative variance across structures than on the absolute magnitude of c-fos expression. We now emphasize the nature of our control group in the discussion:

“Importantly, our control animals were exposed to the conditioning chamber for an equivalent duration without being subjected to shocks, thus encoding and recalling a non-fearful contextual memory.”

1. The author's computation analyses suggested differences in neural networks activated by the retrieval of mild and strong fear memories. The results of computer analysis are interesting. However, it is not clear whether such results are actually occurring in vivo. At this moment, the author's findings are not a conclusion, but rather a suggestion or hypothesis. Therefore, it is also important to conduct interventional experiments to evaluate the validity of the authors' findings. Specifically, the authors' results could be validated by analyzing the effects of inhibition of specific brain regions on mild and strong fear memories retrieval using such as DREADD and other methods. These experiments seem hard, but would greatly improve the quality of the manuscript.

We appreciate the reviewer's perspective and acknowledge the limitations of our current findings. While our data based on c-fos expression suggests functional connections reflective of neural activity during fear memory recall, we agree that it is not possible to deduce causality from this alone. Instead, our study aimed to uncover the network-level distinctions between mild and strong memories, laying the groundwork for subsequent, in-depth investigations of the causal relationships within these identified pathways. We agree that corroborating our findings with interventional experiments, such as using DREADDs, is an important next step. We also agree that such experiments would enhance our study and hope future research will address these points. These points were included in the discussion session:

“To further elucidate the underlying mechanisms of fear memory strength in vivo, understanding the specific roles of individual network elements in fear regulation becomes essential. Future research will be important to probe the causal interplay among distinct nodes and edges, both individually and in combination, in shaping diverse aspects of fear expression.”

**Reviewer #2 (Recommendations For The Authors):**
Methodological detail is lacking:How is the discrimination index calculated?

We have included this information in the methods section: “The generalization index was calculated as Freezing in Test B / (Freezing in Test A + Freezing in Test B).”

A distinction between complete spontaneous recovery (10S group) vs. partial spontaneous recovery (2S group) vs. extinction retention needs to be considered in discussing the extinction data.

Thank you for this suggestion. To address this point, we now include Tukey’s post hoc comparisons between the first and last bins of extinction and the test session. The results show that in the 2S group, freezing during test remained consistent with the levels observed in the final extinction bin and was lower than the levels in the initial extinction bin. Conversely, in the 10S group, freezing levels increased from the final extinction bin to the test, reaching levels comparable to those observed in the initial extinction bin.

Detail regarding the connectivity analyses is missing from the methods. For example the calculation of the r value distractions should be detailed in the methods not just the results, more detail regarding calculations is needed for the degree of centrality, betweenness centrality, nodal efficiency, small world analyses etc.

We appreciate the reviewer’s feedback. We have expanded the description of the connectivity analysis.

Justification for 'excluding edges with r values lower than the average plus one standard deviation of all 292 networks (Figure 4.B; r < 0.61)' is needed.

Thank you for your encouraging us to elaborate on the rationale behind our thresholding method. We acknowledge that there is no consensus in the literature on the optimal thresholding method for functional networks. Our primary objective with thresholding was to retain the most robust connections while minimizing potential noise from weakly correlated regions. Instead of opting for an arbitrary threshold, we determined our cut-off based on the average plus one standard deviation across all networks. Theoretically, this retains approximately the top 16% of connections. Given our 12 regions of interest, this translates to roughly 10 connections per network. This count is sufficient for a nuanced analysis of the network structures and between group comparisons.Importantly, our method inherently accounts for variations in interregional correlations across groups. Groups with a distribution skewed towards higher r values will naturally have more edges, highlighting the enhanced synchronized activity between certain regions. On the other hand, networks with tendencies towards lower r-values will exhibit fewer connections. Thus, our thresholding method is rooted in the data’s distribution and result in networks that reflect the differences across groups.

We added the following sentence to the methods session summarizing this rationale:

“This thresholding approach was used to provide a cut-off based on the data’s inherent distribution, therefore retaining the top edges according to the data variance. “

Line 81 - 'brain areas' is missing after '12'.

Thank you, this is now fixed.

Tile for 2. is somewhat odd. Thought the following may be better, but obviously leaving this up to the author's discretion: 'Commonalities and differences in brain activation induced by recall of mild and strong fear memories'

Thank you for this suggestion. We agree with the title suggested by the reviewer, and it was replaced in the manuscript.

**Reviewer #3 (Recommendations For The Authors):**
1. Previous studies including Karim's lab have shown that protein synthesis in the hippocampus is required for the reconsolidation of contextual fear memory and that the retrieval of contextual fear memory activates gene expression such as c-fos in the hippocampus. However, the authors failed to confirm this observation. This may be due to the small number of rats or some technical problems.

Thank you for this suggestion. As explained above, we believe that this is due to the nature of our control group, which is now highlighted in the discussion section.

1. The author's computation analyses suggested differences in neural networks activated by the retrieval of mild and strong fear memories. The results of computer analysis are interesting. However, it is not clear whether such results are actually occurring in vivo. At this moment, the author's findings are not a conclusion, but rather a suggestion or hypothesis. Therefore, it is also important to conduct interventional experiments to evaluate the validity of the authors' findings. Specifically, the authors' results could be validated by analyzing the effects of inhibition of specific brain regions on mild and strong fear memories retrieval using such as DRRED and other methods. These experiments seem hard, but would greatly improve the quality of the manuscript.

Thank you for your valuable feedback. As explained above, these points are now included in the discussion section.

Minor comments1. cfos should be c-fos or c-Fos.

Thank you for your correction. All instances of ‘cfos’ were replaced by ‘c-fos’.

1. Line 275; "Compared to the to re-exposure to" should be "Compared to the to re-exposure to".

Thank you for your correction. This is now fixed.